# Simulation of Non-Carious Cervical Lesions by Computational Toothbrush Model: A Novel Three-Dimensional Discrete Element Method

**DOI:** 10.3390/s22114183

**Published:** 2022-05-31

**Authors:** Jinsu Nam, Duong Hong Nguyen, Seungjun Lee, Seok-Mo Heo, Junyoung Park

**Affiliations:** 1Department of Mechanical Design Engineering, Kumoh National Institute of Technology, 61, Daehak-Ro, Gumi 39177, Gyeungbuk, Korea; jsnam1005@kumoh.ac.kr (J.N.); 20216074@kumoh.ac.kr (S.L.); 2Techno Vietnam Co., JSC., TSQ Living Area (Euroland), Mo Lao Ward, Ha Dong District, Hanoi City 12110, Vietnam; nguyenhongduong.hut@gmail.com; 3Department of Aeronautic, Mechanical and Electrical Convergence Engineering, Kumoh National Institute of Technology, 61, Daehak-Ro, Gumi 39177, Gyeungbuk, Korea; 4Department of Periodontology, School of Dentistry, Jeonbuk National University, Jeonju 54907, Jeonbuk, Korea

**Keywords:** toothbrush, abrasion, discrete element method, connected particle model, non-carious cervical lesion

## Abstract

Non-carious cervical lesions (NCCLs) are saucer-shaped abrasions of a tooth. NCCLs can form due to various etiologies, including toothbrushing wear, acid erosion, and mechanical stress. Owing to this complex interplay, the mechanism of NCCLs in tooth abrasion has not been established. This study aims to develop a numerical method using a computational toothbrush to simulate NCCLs. The forces acting on the teeth and the amount of abrasion generated were evaluated. The discrete element method using in-house code, connected particle model, and Archard wear model were applied for brushing. In the toothbrush model, 42 acrylic tufts were fixed into a toothbrush head. The teeth models with enamel properties comprised four flat plates and two grooves to simulate the anterior teeth and NCCLs. The brushing speed and depth for one cycle were established as simulation parameters. The force applied within the ununiform plane was concentrated on several bristles as the toothbrush passed through the interproximal space. The brushing force (depth) had a greater effect on tooth abrasion than the brushing speed. Toothbrushing abrasion was mainly concentrated in the interproximal space. Therefore, forceful tooth brushing can cause NCCLs from the interproximal space to the cervical area of the tooth.

## 1. Introduction

Modern manual toothbrushes comprise several components including bristles, a handle, and a head, which holds the bristles and tufts. These standard toothbrushes are widely considered essential for oral health care. If dental plaque, mainly bacterial biofilm, is not properly controlled by toothbrushing, it may cause gingivitis and dental caries, and in severe cases, periodontitis and tooth loss [1,2,3,4,5]. To ensure proper oral hygiene, the mechanical action of toothbrushing is the most common method used for plaque removal [6,7]. Using toothbrushes, however, most people are able to remove less than half of the total plaque present [2,3,8]. Furthermore, traumatic toothbrushing or ‘improper toothbrushing’ may lead to dental and/or gingival abrasion and gingival recession [9,10,11,12].

A non-carious cervical lesion (NCCL) is a saucer-shaped abrasion of a tooth near the cemento-enamel junction (Figure 1). Several studies have shown that the prevalence of tooth abrasion in adults aged 20 to 70 increases from 3% to 17%, respectively [9,10]. Approximately 60% of those with dental erosion also have dentin hypersensitivity as a result of eroded enamel and dentin loss [9,11].

Clinically, NCCLs vary in size and shape, ranging from mild depressions to wedge-shaped or cup-shaped lesions [13]. The formation of NCCLs is attributed to etiologies such as tensile stress [14,15,16], compressive stress [17,18], acid erosion [19,20], stress corrosion [21], abfraction [18,22], occlusal forces, and friction (toothbrush/toothpaste wear) [23,24]. The aforementioned etiologies have varying influences on the shape of NCCLs, (e.g., wedge-, cup-, and smooth-shaped) [24]. The complex interplay of these varying mechanisms complicates the consideration of all NCCLs as a single mechanism.

Generally, three types of occlusal stresses are applied to teeth during mastication and parafunction: tensile, compressive, and shear stresses. The tensile stresses created on the teeth during occlusal loading can contribute to cervical erosive lesions [14,25]. According to certain studies, demineralization of the cervical region is caused by the low pH condition and the piezoelectric effect even under compressive stress, which are concentrated in the cervical region of the tooth [26,27]. Accordingly, tooth fracture may be attributed to such compressive stress. Considering that teeth make contact approximately one million times a year under occlusal stresses with several proteolytic enzymes in the saliva, NCCLs caused by periodic stress cannot be disregarded [28,29]. Numerous studies have investigated brushing abrasion, which is regarded as a common cause of NCCLs. In a recent study, NCCLs were reproduced using common commercial toothpaste and toothbrushes, albeit in vitro [24]. Subjects with good oral hygiene are more prone to NCCLs, and patients who brush their teeth twice daily have a statistically significantly higher incidence of NCCL than those who brush less frequently [30,31]. Therefore, toothbrushing abrasion significantly affects the occurrence of NCCLs.

Dental abrasion might be the result of several factors, one being the interaction between the force applied during toothbrushing and the chemical composition of the toothpaste used [6,9]. Other factors associated with the existence of abrasion and the effectiveness of plaque removal relate to the design of the toothbrush, brush material (length, number of bristles in a tuft, toothbrush head shape and diameter, number and arrangement of tufts, angle between the head and handle and handle design) and the frequency of toothbrush replacement [6,32].

Several theoretical and finite element studies have investigated the bending of single bristles [1,33]. These studies involved both general-purpose toothbrushes and industrial brushes. To date, however, only a limited number of numerical and theoretical studies have been conducted on toothbrushes [34,35]. In part, this is a result of the challenge associated with observing the multiple points of contact between tuft bristles and teeth. Whereas the finite element method can be used to analyze how individual bristles become deformed on teeth contact, it is nearly impossible to analyze the contact between multiple bristles or tufts. In our novel approach, we applied the discrete element method (DEM) to the connected particle model to analyze the multiple bristle-tuft contact points. The DEM is a numerical analysis method typically used in powder analysis, but it can also be used to investigate problems involving multiple contacts [36,37,38].

As mentioned above, NCCLs are closely associated with dental abrasion. However, there are no numerical tools for the analysis of toothbrushing abrasion. Therefore, this study aims to determine the feasibility of a numerical tool for the analysis of abrasion-related lesions and compare the effects of brushing force and brushing speed using the DEM. To the best of the authors’ knowledge, this is the first numerical study to analyze abrasion along tooth surfaces caused by multiple points of contact with bristles.

## 2. Materials and Methods

### 2.1. Computational Model

To simulate the NCCL caused by actual brushing, we need a computational model for the toothbrush and teeth. In addition, a model to calculate the amount of abrasion generated on the tooth by the toothbrush is required.

A computational toothbrush model should have multiple bristles reflecting the shape and material properties of an actual toothbrush. The bristles should be able to bend according to the theory of solid mechanics. In addition, both collisions between bristles and collisions with teeth should be considered. It is nearly impossible to consider both bending and contact in conventional numerical techniques, such as the finite element method. Hence, the bristles are described as connected particles using the DEM, as shown in Figure 2a. In other words, the vertically connected particles depict one bristle or tuft and are bent according to the theory of solid mechanics. A connected particle model based on DEM (numerical technique) was performed using in-house code.

The tooth model consisting of a few planes shown in Figure 2b should also reflect the shape and material properties of the actual tooth. Therefore, two teeth colored brown and an interproximal area colored green are modeled in Figure 2b. The amount of tooth deformation is negligible compared to the toothbrush deformation. Because this is a feasibility study, we model a relatively simple tooth shape.

There is no way to directly “measure” the amount of abrasion on a tooth in a computational model. A portion of the tooth model is not actually removed as much as it is worn out. In other words, the tooth model does not change, and the amount of abrasion is “calculated” through the Archard wear model.

#### 2.1.1. Toothbrush Model

The toothbrush head comprised multiple bristles implanted into the tuft holes. The bristles were typically composed of acrylic or nylon 6/12 filaments and possessed diameters in the range of 0.0375–0.0625 mm. A toothbrush generally possessed 42 tufts with approximately 40 filaments per tuft. To decrease the simulation time, we defined 40 filaments as one beam. The toothbrush had 42 beams; each beam had a diameter of 1.8 mm and a length of 10.8 mm. Each beam was created using eight particles that were 1.8 mm in diameter, and two of these particles were clamped in a similar manner to those in a real toothbrush (Figure 2a).

#### 2.1.2. Tooth Model

The area of the toothbrush head was 27.5 × 10.2 mm. The tooth model used in this study was two flat plates (corresponding to two labial sides of the anterior teeth) connected by a single groove (corresponding to the interproximal space); this model differed from the general shape of a tooth. Because the labial and buccal sides of actual teeth are not flat, it is difficult to determine the vertical and horizontal forces acting on the teeth. For this reason, the four anterior teeth (maxillary and mandibular) and interproximal space were simplified using two planes and one groove, respectively (Figure 2b). The right-hand side of the teeth was 36 mm longer than the left-hand side. An additional plate was attached to the right-hand side of each tooth to reduce the effects of changes in the brushing direction. All teeth measurements were conducted on the planes except for the plate attached to the right-hand side.

The total brushstroke was defined as one forward and one backward movement along the direction parallel to the gingiva, i.e., perpendicular to the interproximal space of the teeth (scrub method). Some previous studies had reported that horizontal brushing is significantly related to NCCL [24,39]. Therefore, in this study, the movement of brushing was purposely simulated in the horizontal direction. The total distance of the two strokes, L, is equal to twice the reciprocal of the tooth model’s length, which is 56 mm.

The material properties of the tooth and toothbrush come from enamel and acrylic (Table 1), respectively, and were obtained by averaging the existing values in the literature [40,41].

#### 2.1.3. Abrasion Model Using Archard Wear Model

The Archard and Oka wear models are mainly used for modeling wear based on particle contact [42,43,44]. The classical Archard model simulates wear by friction, whereas the Oka model imitates wear due to impact. In addition, the Archard model has been utilized in several studies on tooth attrition and orthopedic implants and is thus used in the present study [45,46,47].

The Archard wear model can be described as follows:(1)$$Q=\frac{KWL}{H}$$
where *Q* denotes the total volume of wear produced on the tooth surface, *K* represents the Archard wear constant, *W* denotes the total normal load, *L* is the sliding distance, and *H* denotes the hardness of the material.

By applying this equation to the DEM, we obtain
(2)$$Q=\frac{K}{H}\overset{t}{\underset{0}{\sum }}F_{n}\delta_{t}$$
where *t* denotes the time, *F_n_* is the normal force acting on the tooth surface, and *δ_t_* represents the overlap along the tangential direction. These terms can be easily obtained using the DEM.

When the sphere (the tip of the toothbrush) comes into contact with the plane (the tooth surface), the normal component of the contact force is calculated: ${\vec{F}}_{n}={\vec{f}}_{c}\;\cdot \;\hat{n},$ where ${\vec{f}}_{c}$ is the contact force due to the overlap between the sphere and the plane, and $\hat{n}$ is the unit vector for normal direction. ${\vec{V}}_{c}={\vec{V}}_{cp}-{\vec{V}}_{cs}$ is the relative velocity between the particle and the plane at the point of contact. The sliding distance at the contact point during the time interval *dt* can be expressed as $\delta_{t}=V_{ct}*dt$, where $V_{ct}$ is the tangential component of ${\vec{V}}_{c}$.

When the bristle stiffness of the toothbrush is increased, the force of the bristle tips pressing against the tooth surface increases, that is, *F_n_* increases. Therefore, slightly reducing the stiffness of toothbrush bristles leads to reduced wear of the teeth. However, if the bristle stiffness is lowered and the bristle is excessively bent, the wear distance of the tooth corresponding to *δ_t_* may increase, and wear may increase [9,48]. Therefore, the relationship between bristle stiffness and abrasion (wear) should be considered from a complex viewpoint.

### 2.2. Discrete Element Method (DEM)

The DEM is based on Newtonian mechanics, wherein the forces acting on each particle are numerically integrated to obtain the velocity and angular velocity. Accordingly, the position of each particle is determined via numerical integration. The forces include body (gravity) and surface (contact) forces on the particles.

When particles collide, the contact force on the particle surface is divided into two components: the normal and tangential forces. These forces act along the normal and tangential directions. In the DEM, the contact force is typically calculated using a spring-mass system.

#### 2.2.1. Normal Force

The normal force using a Hertz–Mindlin spring is expressed as follows:(3)$$F_{n}=\frac{4}{3}E^{*}\sqrt{R^{*}\delta_{n}^{3}}-2\sqrt{\frac{5}{6}}\beta \sqrt{S_{n}m^{*}}v_{n}^{rel}$$
(4)$$S_{n}=2E^{*}\sqrt{R^{*}\delta_{n}}$$
(5)$$\beta =\frac{\ln e}{\sqrt{\ln e^{2}+\pi^{2}}}$$
where *E** denotes the equivalent elastic modulus, *R** signifies the equivalent radius, *δ_n_* denotes the normal overlap between particles, *m** indicates the equivalent mass, *v_n_* denotes the relative velocity along the normal direction between contacting particles, and *e* represents the coefficient of restitution.

#### 2.2.2. Tangential Force

The tangential force is expressed as
(6)$$F_{t}=-8G^{*}\sqrt{R^{*}\delta_{n}\delta_{t}}-2\sqrt{\frac{5}{6}}\beta \sqrt{S_{t}m^{*}}v_{t}^{rel}$$
(7)$$S_{t}=8G^{*}\sqrt{R^{*}\delta_{n}}$$
where *G** denotes the equivalent shear modulus, *δ_t_* represents the tangential overlap between particles, and *v_t_* denotes the relative velocity along the tangential direction between the contacting particles. Further details were adopted from the existing literature [48,49,50,51].

### 2.3. Connected Particle Model

To analyze the multiple points of contact with bristle-tufts, a coupled DEM and connected particle model was proposed [38,51,52,53,54,55,56]. Two models are typically used to connect particles to act as one fiber: the bonded particle model and the worm-like chain (WLC) model [39,51,53,54,55,56]. In this study, the WLC model, which is primarily used for deoxyribonucleic acid analysis, was selected [54].

As depicted in Figure 3, in the WLC model, the particles are connected by beam elements. The position and angle of each particle determine the force acting on this beam element. The forces acting on the beam were divided into stretching, bending, and torsion forces.

The stretching force maintains the connection between the particles and the axial beam deformation. The stretching free energy was calculated according to Hooke’s law as follows:(8)$$H^{S}=\frac{1}{2}k_{s}\overset{N}{\underset{i=2}{\sum }}{\left(l_{i}-l_{0}\right)}^{2}$$
where *k_s_* denotes the stretching stiffness, *N* indicates the total number of particles in the beam, *l_i_* represents the distance between the *i*th and (*i −* 1)th particles, and *l*_0_ represents the equilibrium distance between adjacent particles.

The bending free energy can also be derived using the particle position of the beam as follows:(9)$$H^{S}=\frac{k_{B}I}{l_{0}}\overset{N}{\underset{i=2}{\sum }}f_{i}{\left(1-{\hat{t}}_{i+1}\cdot {\hat{t}}_{i}\right)}^{2}$$
where *k_B_* denotes the bending stiffness, *I* represents the areal moment of inertia, and ${\hat{t}}_{i}$ denotes the unit distance vector between adjacent particles.

Similarly, the torsional free energy can be derived using the shear strain of the beam:(10)$$H^{T}=\frac{1}{2}k_{T}\overset{N}{\underset{i=2}{\sum }}{\left(\gamma_{i}-\gamma_{0}\right)}^{2}$$
where *k_T_* denotes the torsional stiffness, *γ_i_* represents the shear strain, and *γ_0_* denotes the equilibrium shear strain.

Each force can easily be obtained by differentiating the free energy derived in Equations (8)–(10). Further details were adopted from the existing literature [55,56].

The brushing speed indicates how fast a person brushes, whereas the brushing depth indicates how forcefully the toothbrush presses against the surface of a tooth. These parameters were selected for the simulation. The brushing depth was determined as the depth at which the initial position of the bristle-tip particles descended below the tooth surface (Figure 4). In this study, three brushing depths, *d_z_*, were selected based on the radius of the bristles (r, r/2, and r/4). The brushing speeds were 0.05, 0.10, and 0.15 m/s. The diameter of the bristles was 1.8 mm, corresponding to an *r* value of 0.9 mm.

### 2.4. Model Validation

The DEM used for considering multiple contact points was not validated in this study, but it may be found in numerous previous studies. However, the WLC model is validated using the cantilever beam deflection test, as this is the first time that it has been applied to toothbrushes (Figure 5a).

As depicted in Figure 5b,c, the overall trend of the brush beam is in good agreement with that of the nonlinear theoretical equation. Even at 30 N, which is the greatest applied force considered in this study, the error of 3.3% (approximate) is negligible.

## 3. Results

### 3.1. Dynamic Behavior of Toothbrush

As shown in Figure 6, when the toothbrush moves forward, the bristles are deflected backward and vice versa. In addition, some bristles are bunched together while others are spread apart in the spaces between the teeth (i.e., interproximal space). The arrangement of the bristles was quite complex as they were located in the interproximal space at the positions of L/5 (Figure 6a) and 4L/5 (Figure 6d). When the bristles pass through the interproximal space, they are bent and forcefully released, thereby hitting the side surface of the teeth. Accordingly, considerable abrasion is highly likely to occur in this section, which is strongly affected by the movement of the bristles.

### 3.2. Contour Plots of Forces Acting on the Toothbrush

During the brushing process, the bristles or tufts exert an impact on the plaque, removing it from the surface of the teeth; however, they also promote abrasion. Therefore, it is important to analyze the force exerted on the tooth surface by the bristles. In this study, the force acting on the tip of the bristles was considered by dividing this force into the normal force acting in the direction perpendicular to the surface and the tangential force acting along the direction in contact with the surface, as shown in Figure 7a,b. The bristles were located in the interproximal space at the positions of L/5 and 4L/5. The maximum magnitude of the normal and contact force in the interproximal space, approximately 0.05 N, was 10 times greater than the average force magnitude, approximately 0.005 N.

### 3.3. Abrasion Distribution on the Surface of Teeth

The amount of wear was calculated using the Archard wear model. Because chemical and biological erosions were not considered in this study, the amount of wear corresponds directly to the amount of abrasion. Figure 8 illustrates the spatial distribution of the amount of abrasion that occurs over one cycle of toothbrushing according to varied brushing speeds and depths.

## 4. Discussion

Many studies have been conducted to investigate the relationship between the stiffness of bristles and plaque removal/abrasion [9,48,58,59]. Although tooth enamel and dentin exhibit high hardness, they inevitably encounter problems due to abrasion. Contrary to the general belief, when a manual toothbrush with various bristle stiffnesses was tested in terms of abrasion on eroded and mechanically sound human dentin, it was found that bristles with a soft stiffness exhibited high dentin abrasion [9]. In addition, high abrasion occurs when soft bristles are applied to normal and demineralized human enamel [48]. Conversely, when the stiffness of a toothbrush increases, damage to the gingival tissue, (i.e., gingival recession) occurs while plaque removal increases [58,59]. Without abrasive toothpaste or abrasive powder, the number of gingival recessions caused by hard brushes increases by about four times that of those caused by soft brushes [59]. On the other hand, the plaque score using the hard brush increases by 20% in comparison with using the soft brush [58]. Using the Quigley–Hein index modified by Turesky, (i.e., the plaque index), it has been demonstrated that the plaque removal rate differs among three types of manual toothbrushes [6]. Similarly, an attempt was previously made to analyze the effects of the load and toothpaste on the vibration characteristics of powered toothbrushes using the noncontact technique of scanning-laser vibrometry [60]. Several studies have also experimentally observed the effect of bristle stiffness and bristle-tip geometry on the behavior of abrasive particles in toothpaste [61,62]. Compared to soft brushes, hard brushes may act as a barrier to prevent particles from escaping, so the abrasive particles stay on the toothbrush longer. In addition, the direction of the brush, the number of filaments in the tuft, and the spacing of the tufts affect the entry and exit of particles. Nevertheless, these studies were conducted based on experimentation. Although not mentioned directly in other papers, it is possible that bristles wear may be related to NCCL. If the bristle length made of the same material is shortened by wear, the stiffness increases, leading to the occurrence of NCCL.

In general, plaque removal is mainly performed by toothbrushing. However, traumatic or improper toothbrushing can bring forth dental abrasion that may lead to NCCLs [9,10,11]. So far, proper toothbrushing mechanisms or the exact causes of NCCLs have not been established. To overcome this limitation, it is important to understand the operating mechanisms of toothbrushing considering parameters such as the brushing speed and the force applied by the toothbrush bristles on the teeth. Therefore, numerical studies are essential for this purpose.

In the first step of the toothbrushing simulation, the deformation of the model was validated. Because the bristles have three deformation modes, bending, tension, and torsion, three validations are needed. However, because the tension and torsion errors observed in previous studies were relatively negligible considering connected particle models and given that the major force generated during brushing is bending, the developed model was validated only for bending [37,53]. The bending of a bristle or tuft during brushing can be expressed by the deflection of a cantilever in solid mechanics [1]. Therefore, we used one beam composed of eight particles to describe one tuft, as illustrated in Figure 5a, and compared the results of these simulations with those obtained via theoretical solid mechanics calculations. The error between these two methods is depicted in Figure 5b,c. The measured error at the maximum force, 30 N, was 3.3%. Considering that the force applied to teeth is generally less than 10 N, it can be confirmed that the deflection of the toothbrush bristles will have a negligible error in the general brushing force [9].

In the second step, the spatial distribution of the normal and tangential forces was analyzed, as they are directly related to the amount of abrasion or plaque removal. Figure 7a,b show the instantaneous normal and tangential forces applied to the teeth, respectively. Figure 7a depicts the results obtained by setting the brushing speed to 0.1 m/s and brushing depth to three values: r (radius of the bristles), r/2, and r/4. At position L/5 where the bristles were in the interproximal space during the forward stroke, the normal and tangential forces exhibited neither a clear difference nor a tendency to increase or decrease. The normal and tangential forces both increased, however, as the brushing depth decreased (from *d_z_* = *r* to *d_z_* = *r*/4) at position 4L/5, when the toothbrush was in the interproximal space during the backward stroke. In particular, the forces were concentrated on a few specific bristles rather than the overall bristles. Figure 7b depicts the results obtained at a toothbrush penetration depth of *d_z_* = *r* at three brushing speeds of 0.05, 0.10, and 0.15 m/s. Similar to Figure 7a, the forces in Figure 7b are generally concentrated on a few specific bristles. While the relationship between the magnitude of the force and speed is unclear at position L/5, the magnitude of the force decreased as the speed increased at position 4L/5.

Overall, at position L/5, the normal and tangential forces were less affected by the brushing speed and depth, while at position 4L/5, they were affected considerably. The tangential force exhibited a greater overall value in comparison with that of the normal force. The magnitude of the forces changed considerably when the brushing speed was fixed and brushing depth was varied, compared with the case in which the brushing depth was fixed and brushing speed was varied. This shows that the brushing depth had a greater effect on the force magnitude than the brushing speed.

Lastly, as depicted in Figure 8, the amount of abrasion increases as the brushing speed decreases and brushing depth increases. In particular, the abrasion becomes severe on both sides of the interproximal space under these conditions as the bristles impact the side of the tooth, as shown in the red box in Figure 8. Therefore, it is inferred that the NCCL associated with dental abrasion initiates from the side (interproximal space) and spreads to the center (tooth area). As explained in the previous section, the effect of the brushing depth is larger than that of the brushing speed; therefore, the difference between *d_z_* = *r* and *d_z_* = *r*/4 is larger than that between the speed settings of 0.05 and 0.15 m/s. Overall, the abrasion exhibits a top-bottom symmetrical distribution. This is because the arrangement of the bristles is symmetrical and the arrangement of the anterior teeth of the maxilla and mandible are perfectly identical, thereby yielding nearly identical shapes with a symmetrical distribution.

As mentioned earlier, we modeled the scrub method in this study. Therefore, all results are only applicable for the scrub method, and results for other methods, such as the Bass and Roll methods, may differ [3].

In previous studies, the implementation of numerical approaches for the analysis of brushing abrasion was inhibited by the complexity of simultaneously analyzing deformation and multiple points of contact [33,34,35,63]. Conventional approaches based on the finite element method allowed for the accurate observation of the deformation mechanism of bristles. However, the analysis of the multiple points of contact with the bristles was not possible [33,37,39]. We simulated the toothbrushing process based on DEM, accounting for multiple points of contact.

A limitation of this study is that we used a flat tooth model and treated the filaments tuft of the toothbrush as one tuft. Therefore, it is essential to improve the tooth model and toothbrush model to a more detailed one in the future.

## 5. Conclusions

This study was the first to successfully implement a novel three-dimensional DEM analysis on a computational toothbrush model that allowed the numerical analysis of the extent of toothbrushing abrasion such as NCCLs. The force applied during toothbrushing (the depth of brushing) has a greater effect on tooth abrasion than the speed of brushing. In addition, toothbrushing abrasion was concentrated in the interdental space. Therefore, forceful toothbrushing in our daily lives can cause NCCLs to develop from the interproximal space to the cervical area of a tooth. Accordingly, as a next strategy, we plan to analyze wear by modeling and applying each bristle to real human teeth. In the future, it would be meaningful to investigate the relationship between tooth wear and interdental embrasures by applying the DEM model in a multi-centered clinical trial with a large number of patients.

## Figures and Tables

**Figure 1 sensors-22-04183-f001:**
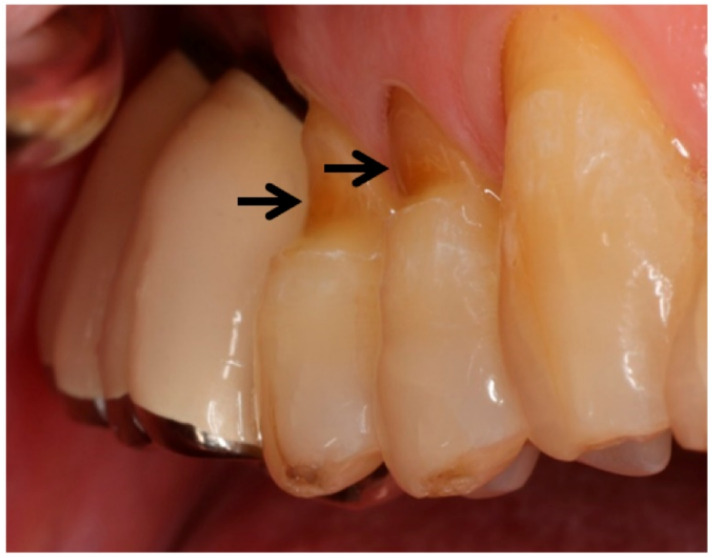
Clinical case of non-carious cervical lesions of teeth (indicated by arrows) caused by traumatic toothbrushing.

**Figure 2 sensors-22-04183-f002:**
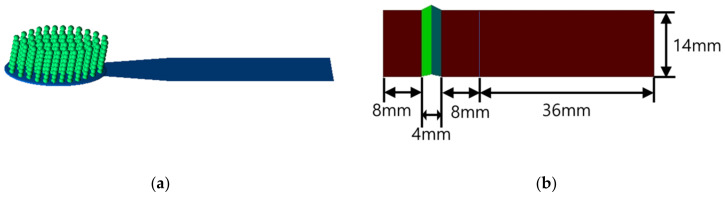
Computational model for (**a**) toothbrush and (**b**) tooth.

**Figure 3 sensors-22-04183-f003:**
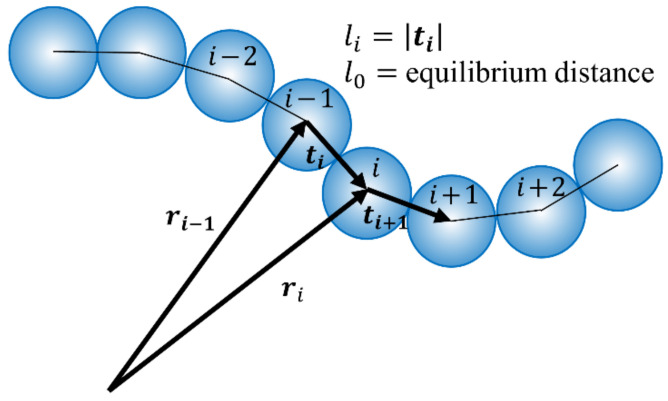
Schematic of worm-like chain (WLC) model used as a connected particle model.

**Figure 4 sensors-22-04183-f004:**
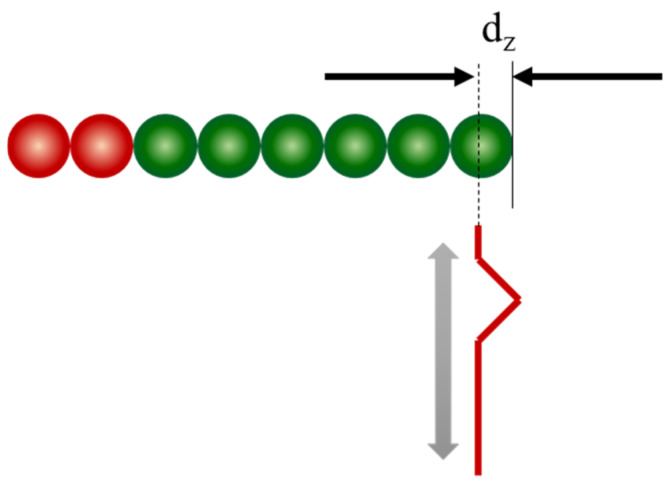
Schematic of brushing depth.

**Figure 5 sensors-22-04183-f005:**
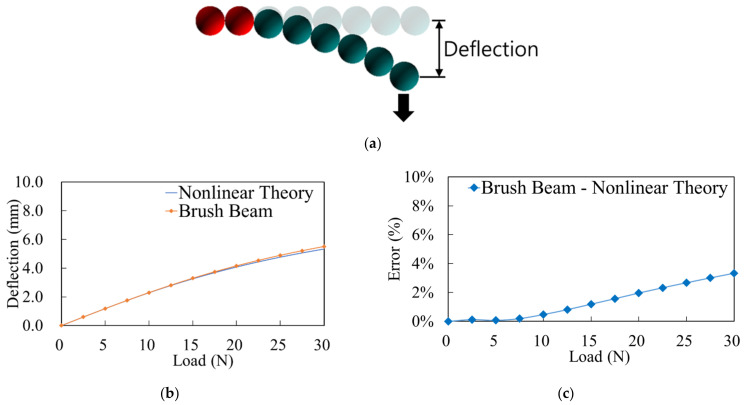
Comparison between nonlinear theory and discrete element method (DEM) results; (**a**) cantilever beam deflection represented by the WLC model; (**b**) deflection; (**c**) estimated error.

**Figure 6 sensors-22-04183-f006:**
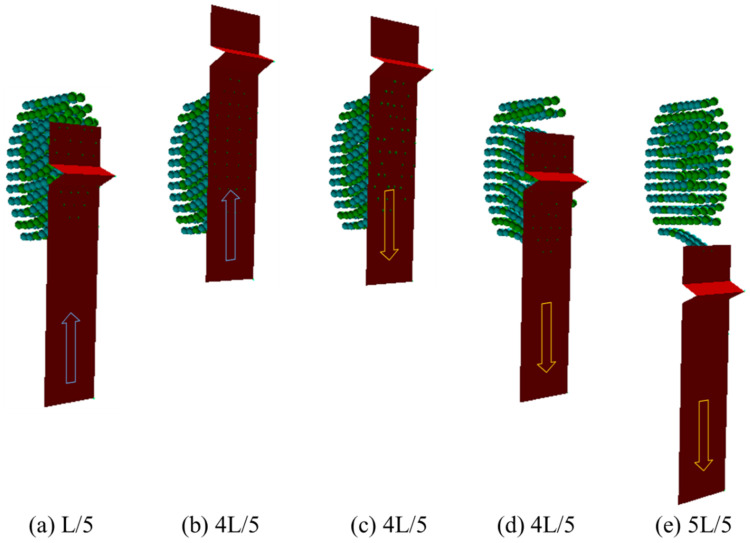
Dynamic behavior of toothbrush at the given positions in the case of *d_z_* = *r* and *V* = 0.05 m/s [57].

**Figure 7 sensors-22-04183-f007:**
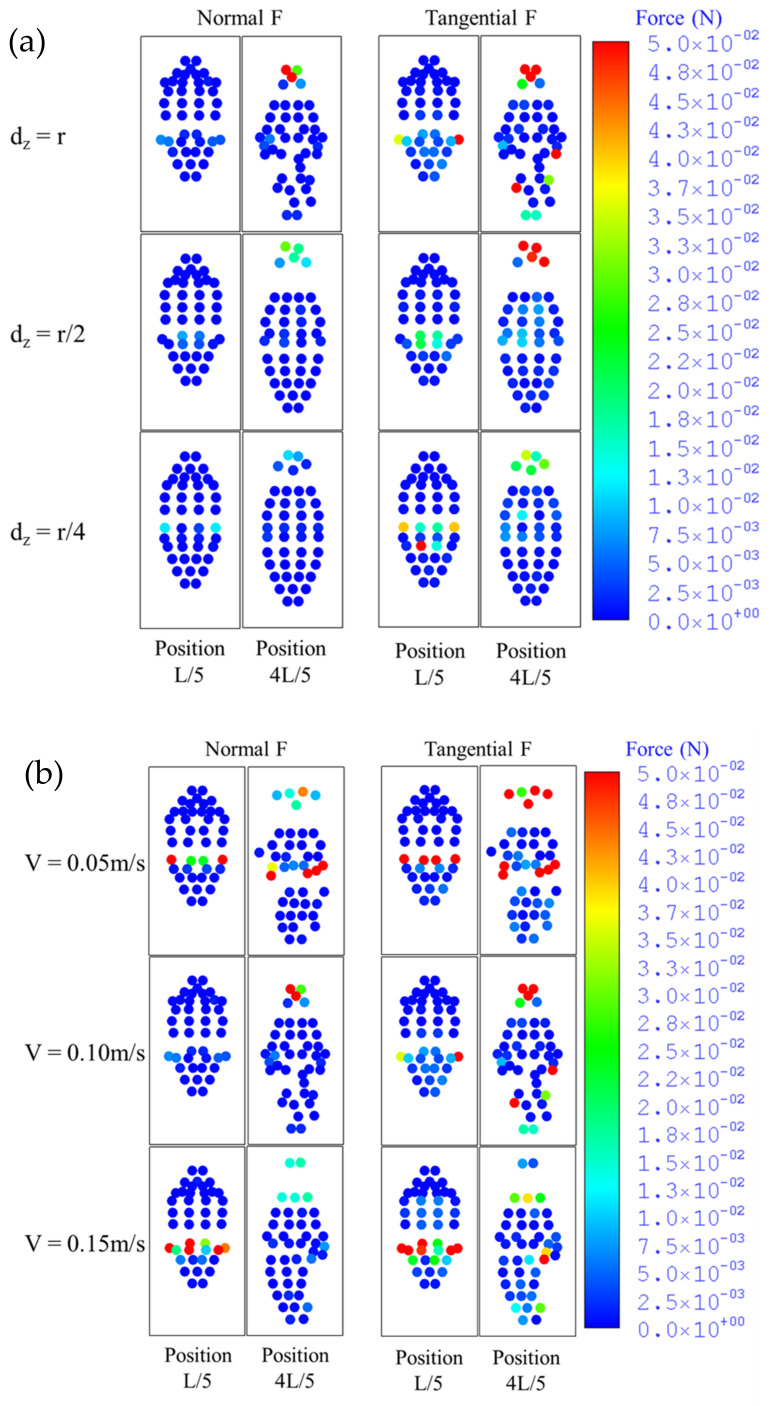
Contour plots of forces acting on toothbrush for (**a**) various toothbrush penetration depths at fixed toothbrush speeds (*V* = 0.1 m/s) and (**b**) various toothbrush speeds at fixed toothbrush penetration depths (*d_z_* = *r*) [57].

**Figure 8 sensors-22-04183-f008:**
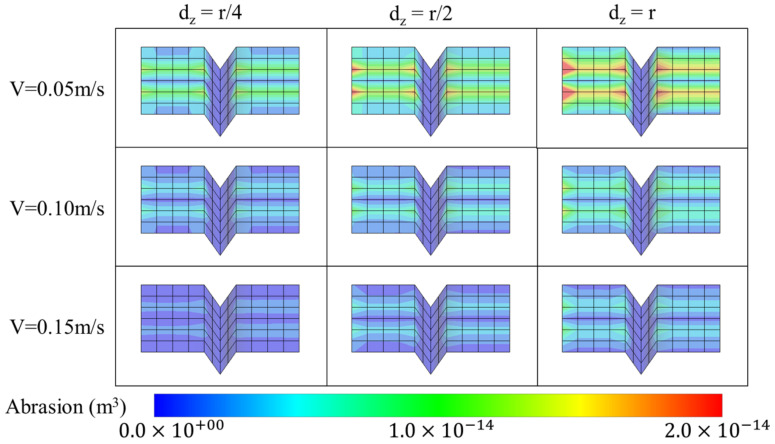
Abrasion distribution on tooth surfaces under various conditions.

**Table 1 sensors-22-04183-t001:** Material properties.

Material		Acrylic Brush	Tooth Enamel	Unit
Elastic modulus		3.4	60.0	GPa
Density		1185	3000	kg/m^3^
Hardness		0.22	0.92	GPa
Poisson’s ratio		0.3	0.3	
Coefficient of friction	Acrylic	0.18	0.18	
	Enamel	0.18	N/A	

## Data Availability

The data presented in this study are available on request from the corresponding author.

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
