# Peer review of "Simulation of Non-Carious Cervical Lesions by Computational Toothbrush Model: A Novel Three-Dimensional Discrete Element Method"

_sensors, 2022, doi:10.3390/s22114183_

Round 1

Reviewer 1 Report

Dear authors,

Thank you for the opportunity to review your article. The present manuscript approaches an important issue for the dental community, that is the mechanism of formation of non-caries cervical lesions (NCCL). The paper is well organized and presents interesting results regarding the usability of a numerical computational model to simulate and investigate the action of toothbrushing in the mechanism of formation of NCCL, which, in my opinion is the novelty of the work. The employed methods were detailly described and the discussion is scientifically sound and supported by proper references. My only question is with some conceptual mistakes I saw in the manuscript (for example: page 10, line 297:  “… to the expected lifespan of biomaterials such as teeth.”  Certainly, teeth are not a biomaterial). Therefore, I recommend submitting the manuscript to a careful revision in order to eliminate this in other mistakes and also to improve its English style. 

Reviewer 2 Report

The authors aimed to develop a numerical method using a computational toothbrush to 19 simulate non-carious cervical lesions (NCCLs). I found the paper well conducted and interesting since the study is related to clinical outcomes. Here some minor suggestions to improve the quality of the manuscript:

ABSTRACT

  • Lines 28-29: Remove “(i.e. the space between adjacent teeth)” and correct with “… in the interproximal dental space”.

INTRODUCTION

  • Lines 35-42: better specify the consequences of bad plaque removal (gingivititis, dental caries and periodontitis) and improper toothbrushing (gingival recession) with some new references. This is not necessary for NCCLs (which are discussed immediately after).
  • Lines 72-75: add a consideration on the need and frequency of toothbrush replacement (wear of the bristles).
  • Figure 1: specify the etiology of the NCCLs.

MATERIALS AND METHODS

  • Paragraph 2.1.2: Why did you choose an horizontal movement parallel to the gingiva, movement that dentists usually advise against for patients? Did you want to purposely simulate a movement that is not ideal but frequently adopted by patients? Please better specify in the text.

DISCUSSION

  • Lines 303-304 (Ref. n 54-55): could you better specify the evaluated clinical outcomes for “damage to the gingival tissue” and “plaque removal”?
  • Lines 309-310 (Ref. n 57-58): please specify the results of these studies.
  • Lines 295-311: Could you discuss something about the wear of the bristles related to NCCLs?
  • Lines 312-314: what is exactly intended for “improper toothbrushing” in ref. n 7-9?
  • Lines 314-315: missing references.
  • Better specify at the end of the discussion the limitation of the study (for example regarding the toothbrush and the tooth models).

Reviewer 3 Report

Well written and scientifically sound manuscript.
